# Morphological, Material, and Optical Properties of ZnO/ZnS/CNTs Nanocomposites on SiO_2_ Substrate

**DOI:** 10.3390/nano10081521

**Published:** 2020-08-04

**Authors:** Yu Sheng Tsai, Xin Dai Lin, Wei Lun Chan, Shang Che Tsai, Wei Jen Liao, Yew Chung Sermon Wu, Hsiang Chen

**Affiliations:** 1Department of Materials Science and Engineering, College of Engineer, National Chiao Tung University, Hsinchu 30050, Taiwan; s0781509.mse07g@nctu.edu.tw (Y.S.T.); sermonwu@faculty.nctu.edu.tw (Y.C.S.W.); 2Department of Applied Materials and Optoelectronic Engineering, College of Science and Technology, National Chi Nan University, Puli 54561, Taiwan; s105328044@mail1.ncnu.edu.tw (X.D.L.); s106328031@mail1.ncnu.edu.tw (W.L.C.); s106328033@mail1.ncnu.edu.tw (S.C.T.); liauweijen@gmail.com (W.J.L.)

**Keywords:** ZnO/ZnS coreshell NRs, CNTs, optical properties, I–V curves, cross section

## Abstract

Ultraviolet A light (UV-A, 320–400 nm), which is unblockable by sunscreen, requires careful detection for disease avoidance. In this study, we propose a novel photosensing device capable of detecting UV-A. Cancer-causing UV light can be simultaneously monitored with tiny rapid response sensors for a high carrier transition speed. In our research, a multifunctional ZnO/ZnS nanomaterial hybrid-sprinkled carbon nanotube (CNT) was created for the purpose of fabricating a multipurpose, semiconductorbased application. For our research, ZnO nanorods (NRs) were grown by using a facile hydrothermal method on SiO_2_ substrate, then vulcanized to form ZnO/ZnS coreshell nanorods, which were sprinkled with carbon nanotubes (CNTs). Results indicate that SiO2/ZnO/ZnS/CNT structures exhibited a stronger conducting current with and without light than those samples without CNTs. Multiple material characterizations of the nanostructures, including of atomic force microscopy (AFM) surface morphology evaluation, scanning electron microscopy (SEM), and transmission electron microscopy (TEM) indicate that CNTs could be successfully spread on top of the ZnO/ZnS coreshell structures. Furthermore, chemical binding properties, material crystallinity, and optical properties were examined by X-ray diffraction (XRD), energy dispersive spectroscopy (EDS), and photoluminescence (PL). Owing to their compact size, simple fabrication, and low cost, ZnO/ZnS coreshell NRs/CNT/SiO_2_-based nanocomposites are promising for future industrial optoelectronic applications.

## 1. Introduction

Among wide-band gap semiconductor materials, ZnO/ZnS coreshell structures have been regarded as one of the most promising photodetection candidates owing to their excellent optical properties and high electron mobility [1]. Specifically, heterojunction of a ZnO/ZnS double layer can convert UV light into conducting carriers capable of detecting UV photons. Toward this goal, coreshell heterostructures such as ZnO-TiO_2_ [2], ZnO-SnO_2_ [3], ZnO-ZnSe [4], and ZnO-ZnS [5] have been successfully constructed in the past. In addition, 1D ZnO/ZnS nanostructures can facilitate the rapid transfer of photogenerated electron-hole pairs [6,7]. Therefore, integration of ZnO/ZnS materials into optical or gas sensing chips can usually achieve high sensing capability [8,9]. Recently, the incorporation of carbon nanotubes (CNTs) into ZnO nanorods has been demonstrated to boost ZnO nanorod (NR)-based gas sensing performance due to the speeding up the carrier transition near the ZnO NR/substrate interface. In addition, the ultrahigh conductivity of CNTs may enhance the carrier transition speed in sensing devices [10,11]. In this study, CNTs were added to ZnO/ZnS coreshell structures. Multiple morphological and material characterizations reveal that CNTs could be well attached to the top of ZnO/ZnS coreshell NRs, and fine interfacial ZnO/ZnS/CNT nanostructures were observed by transmission electron microscopy (TEM) images. Finally, dark current and light current current–voltage (I–V) measurements for ZnO/ZnS coreshell structures with and without incorporation of CNTs reveal that the addition of CNTs on ZnO/ZnS NRs could boost both dark and light currents, and therefore enhance photosensing properties [12,13]. ZnO/ZnS/CNT nanocomposites are promising for future photodetection and optical-sensing applications [14,15].

## 2. Materials and Methods

To synthesize grain-like ZnS on ZnO structures, wafers were first cut into 2 × 2 cm substrates. Then, the silicon substrates were sequentially cleansed with ethanol, acetone, and isopropyl alcohol were received from Shimakyu’s Pure Chemical (Osaka, Japan). The ZnO seed layer was spin coated onto the substrates with a lab-made solution. Then, ZnO nanorods were grown hydrothermally on the substrates at 80 °C in a solution containing 0.05 M zinc nitrate hexahydrate and 0.07 M hexamethylenetetramine, which was purchased from Alfa Aesar (Ward Hill, MA, USA), for one hour. After the cleaning and drying processes, the substrates synthesized with ZnO nanorods were placed in a solution containing 0.05 M sulfide nanohydrate, which was purchased from Acros (Pittsburgh, PA, USA), at 70 °C for 0, 15, and 30 min to form ZnO/ZnS core–shell NRs structures. Finally, the CNTs were sprinkled onto the ZnO/ZnS core–shell nanorod structures.

### 2.1. Preparation of the Seed Layer for ZnO by Sol–Gel Method and Spin Coating

A total of 0.66 g of Zinc acetate (Zn(CH_3_COO)_2_, 0.05 M), which was purchased from Acros was added into 60 mL of ethanol (C_2_H_5_OH). Two drops of monoethanolamine (MEA), which was purchased from Acros were then dripped into the solution as a stabilizer. This was placed on a SiO_2_ substrate (with electrode and without electrode) into a spin coater, and two to three drops of spin-coating aqueous solution were dripped onto the surface of the substrate. In order to uniformly spin coat the seed layer solution, the substrate was first spun at 500 rpm for the first 5 s, then 3000 rpm for the final 30 s. After spin coating, the substrate was placed at the top of a 130 °C-heated proportional–integral–derivative (PID) control heating plate, made by Shin Kwang (Taiwan), for 5 min, and the above steps were repeated 5 times.

### 2.2. Synthesis of Array Nanorods by Hydrothermal Method

Dissolved zinc ionic compound in alkaline aqueous solution OH^−^ (released by Hexamethylenetetramine, HMTA) was formed into compounds with Zn^2+^. Then, a ZnO core was grown in the direction of the nonpolar side, forced to an anisotropy growth in the c-axis direction, and finally synthesized into one-dimensional NRs on the substrate. The temperature and growth time for hydrothermally grown ZnO NRs were 80 °C and 60 min, respectively. The chemical reaction formulas are as follows:C_6_H_12_N_4_ + 6H_2_O → 4NH_3_ + 6HCHO
NH_3_ + H_2_O → NH_4_^+^+ OH^−^
Zn^2+^ + 2OH^−^ → Zn(OH)_2_
Zn(OH)_2_ → ZnO + H_2_O

To synthesize the ZnS shell, a 2-stage hydrothermal method was required, with ZnO NRs as the reaction template, in order for the anion exchange reactions to proceed. The growth temperature and time for the hydrothermally grown ZnS shell were 70 °C and 15 and 30 min, respectively. The chemical reaction formula is as follows:ZnO + Na_2_S + H_2_O → ZnS + 2NaOH

### 2.3. Doping Carbon Nanotube (CNT)

Through a quantitative dropper, CNTs of 10 microliters could be uniformly dripped on the substrate, and the substrate was dried using a 70 °C heating plate. An illustration of the ZnO/ZnS/CNT fabrication process is shown in Figure 1.

## 3. Results and Discussion

The surface morphologies of CNT/ZnO/ZnS nanocomposites can be observed by field emission scanning electron microscope (FE-SEM), which was manufactured by FEI (Hillsboro, OR, USA) [16,17]. Sprinkled CNT dripped on ZnO/ZnS core–shell nanostructures with vulcanization times of 15 and 30 min is shown in Figure 2a,b and no significant difference on the surface morphology can be noticed. Furthermore, SEM cross-section images of the sample cut by focus ion beam (FIB) reveal that CNTs were successfully sprinkled on top for CNT/ZnO/ZnS with vulcanization times of 15 and 30 min, as shown in Figure 2c,d. In addition, the cross-section image clearly shows the dimensions of the CNTs, ZnO, and ZnS. Figure 2c shows the thickness of CNTs was around 190 nm, the width of the ZnO/ZnS NRs was around 40 nm, and the height of the NRs was around 500 nm, while Figure 2d shows the thickness of CNTs was around 300 nm, the width of the NRs was around 40 nm, and the height of the NRs was around 500 nm [18,19].

In order to explore the crystalline phase of the ZnO/ZnS core–shell samples, X-ray diffraction (XRD) patterns, which was made by Bruker (Billerica, MA, USA), were studied, as shown in Figure 3a. All the diffraction peaks are compared with the information table of the Joint Commitee on Powder Diffraction Standards (JCPDS). For ZnO NRs without vulcanization, only ZnO phases were observed (JCPDS.792205), and the diffraction peaks of ZnO in 2*θ* were 31.7° (100), 34.5° (002), 36.3° (101), 56.7° (110), 63.0° (103), and 68.0° (112). In this study, the (002) lattice plane possessed the strongest diffraction peak, indicating that the NRs were grown along the c-axis [20], as shown in Figure 3c. As the vulcanization time increased to 15 and 30 min, a cubic sphalerite (111) ZnS phase (JCPDS.790043) appeared [21], and a diffraction peak of (111) ZnS was found around 28.8° [22,23], showing that vulcanization particles indeed covered up the surface of the ZnO NRs. In addition, before the CNTs were sprinkled, CNT phases could not be found. However, after sprinkled CNTs were added, a CNT diffraction peak at 2*θ* = 26.18° could be observed. Next, we will focus on the size-strain analysis for microstructure. After comparing each Full width at half maximum (FWHM) of the samples, we could tell that after the vulcanization process the value of FWHM of the ZnO (002) peak had decreased, which also meant that the grain size became bigger and with longer vulcanization time the more expansion it got. As for the strain analysis, by observing the strongest peak, that of the ZnO (002) lattice plane of the phase, we found out that the position of peak ZnO (002) had slightly shifted to higher 2*θ* values after vulcanization, indicating the interplanar of ZnO had expanded. In contrast, the interplanar spacing of ZnS had decreased.

Furthermore, ZnO/ZnS core–shell structures were characterized using photoluminescence (PL) spectroscopy to observe crystal defects and optical properties [24]. PL was made by HORIBA Jobin Yvon SAS (Kyoto, Japan). Two wavelengths discovered by PL represent different meanings for their emission peaks, as shown in Figure 3b. A near band edge (NBE) peak [10] at 380 nm, caused by energy absorption, was observed. Furthermore, NBE peaks with vulcanization exhibited a red-shift phenomenon. In addition, the CNT-sprinkled samples had a more obvious red-shift phenomenon, and an oxygen defect peak caused by a crystal defect at 560 nm was discovered. According to previous research [25], photosensing ability is enhanced with a higher defect peak value. After evaluation, we discovered that the CNT-sprinkled sample had stronger defect luminescence; thus, it can be expected to have the best photoelectric characteristics. Moreover, the results correspond with the light current–voltage measurements.

Atomic force microscopy (AFM) measurements were used to study the surface roughness of the ZnO/ZnS/CNT structures with vulcanization times of 15 and 30 min, as shown in Figure 4. AFM was manufactured by Bruker (Billerica, MA, USA). The samples had a surface roughness (Ra) of 0.0326 and 0.0358 μm, respectively. Surface roughness (Ra) means the absolute value of the arithmetic average of profile roughness measurement in AFM. Root mean square deviation (Rq) means the height deviation taken from the mean image data plane. The roughness was consistent with the cross-section images of Figure 2c,d, and results indicate the thicker CNT would result in higher roughness. In the measurement, the Ra of the ZnO/ZnS 15 min/CNT from the lowest to the highest peak of the measured surface (approximately 200 nm) was similar to the thickness of the CNTs of the samples cut by FIB, as shown in Figure 4a. Likewise, the Ra of the ZnO/ZnS 30 min/CNT, shown in Figure 4b, corresponded with the thickness of the CNT layer in the cross-section image. The AFM measurements also revealed sprinkled CNTs on the top of the NRs by speculating the height difference in the Ra.

To examine the detailed nanostructures of the nanocomposites, TEM was performed. The manufacturer of TEM was Jeol (Akishima, Japan). Figure 5a shows a TEM image of ZnO/ZnS 15 min without CNTs, and a red circle is used to specify the location of an HRTEM image in Figure 5b. The high-resolution TEM clearly shows a perfectly distinguishable border separating the two different materials [26], proving that we have successfully synthesized a ZnS shell, which covered ZnO nanorods to form a core–shell nanostructure. Figure 5c shows a TEM image of ZnO/ZnS with a vulcanization time of 15 min with sprinkled CNTs on top. Moreover, images with gray and red circles were used to specify different locations on the HRTEM images [27], as shown in Figure 5d,e. Figure 5d shows an HRTEM image of a CNT on ZnO/ZnS surface, and Figure 5e shows the lattice plane of ZnO (002) and ZnS (111) without CNT. The interplanar spacing of ZnO = 0.246 nm and ZnS = 0.334 nm was measured using digital micrograph software [28,29]. Results further confirmed the presence of CNTs on the top of the ZnO/ZnS core shell, and not at the bottom [30].

Finally, to study the elemental compositions of the ZnO/ZnS/CNT structures, a sample with sprinkled CNTs and a vulcanization time of 30 min was selected to perform energy dispersive X-ray spectroscopy (EDS) analysis, as shown in Figure 6 [21]. EDS was made by FEI (Hillsboro, OR, USA). Table 1 showed a sorted data of EDS analysis of the ZnO/ZnS core–shell/CNT nanostructures. Our lab has previously released a paper of the research into the ZnO/ZnS nanostructure [27], so in this experiment we focused our research on the characteristics of ZnO/ZnS/CNT nanostructures. Apparently, higher carbon concentration and lower Zn concentration occurred on the top of the structure where CNTs did not come into contact with ZnO (spectrum 1, 3 had a low percentage of Zn), while lower carbon and higher Zn concentration occurred on the bottom where CNT had come into contact with ZnO (spectrum 2 had a high percentage of Zn), as shown in the EDS table. Moreover, the results correspond with SEM images and cross sections, as shown in Figure 2a,b and Figure 2c,d, respectively. On the top of the ZnO/ZnS NRs, the carbon ratio is very high, so it can be inferred that the CNTs are successfully sprinkled on the top.

The dark current–voltage measurements as shown in Figure 7a, show that the I–V with CNTs sprinkled on the top had stronger current, owing to the high conductivity of the CNTs. We measured dark current and light current by Agilent (Santa Clara, CA, USA).The light current–voltage measurements also indicate that the samples with CNTs had stronger current, but the increased current rate caused by the light was similar or slightly inferior to the samples without CNTs. A comparison of dark current and light current was shown in Table 2. This result may be caused by the blocking of the light by the sprinkled CNTs on the top [16,17]. Generally speaking, CNTs could greatly enhance conductivity and current [31,32].

## 4. Conclusions

In this research, ZnO/ZnS core–shell NR/CNTs were fabricated on top of interdigitated electrode/SiO_2_ substrates. Multiple material analyses including FE-SEM, XRD, PL, AFM, and TEM were performed. Material and morphological analyses indicate that CNTs were attached on top of the ZnO/ZnS nanostructures. Electrical measurements reveal that ZnO/ZnS core–shell NR/CNTs had stronger conductivity and current than the samples without CNTs. On the other hand, CNTs on top might cover ZnO/ZnS structures from being exposed to a small amount of light. Owing to low cost, simple fabrication processing, and small size, ZnO/ZnS core–shell NRs/CNTs nanomaterials show promise for future industrial optoelectronic applications.

## Figures and Tables

**Figure 1 nanomaterials-10-01521-f001:**
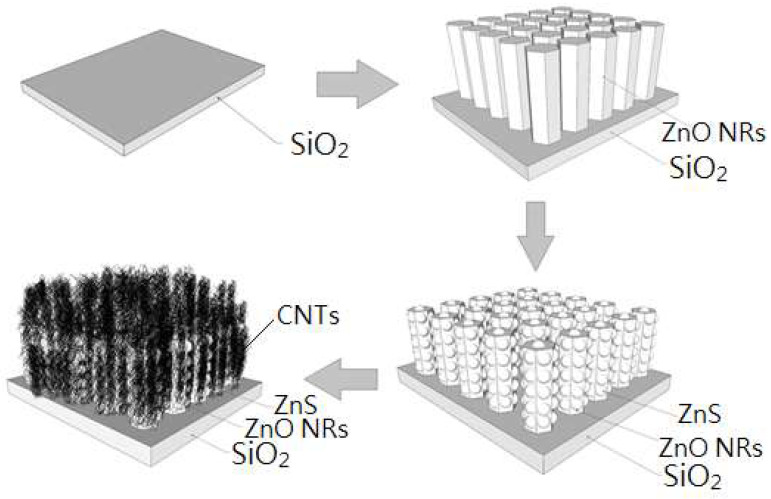
Schematic diagram of fabrication processes of ZnO/ZnS core–shell nanostructure/carbon nanotubes (CNTs) on SiO_2_ substrate.

**Figure 2 nanomaterials-10-01521-f002:**
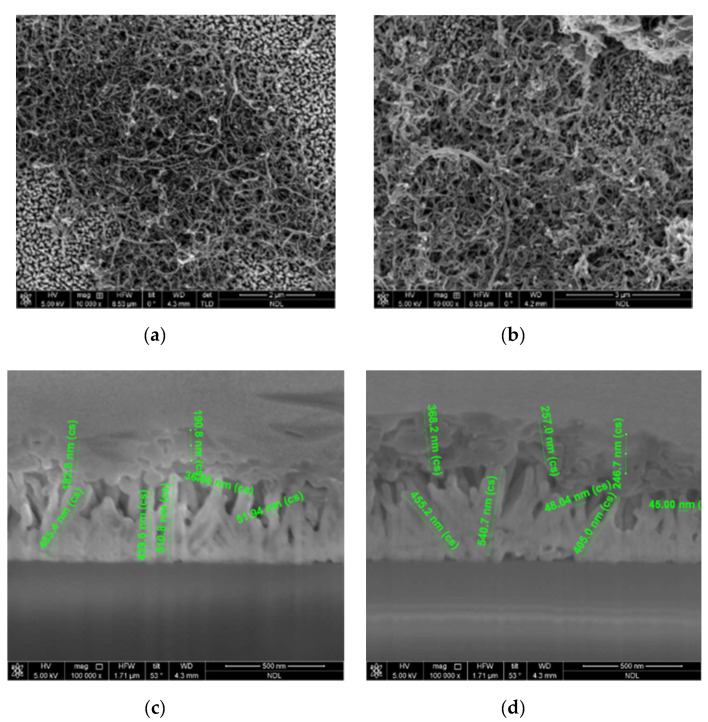
(**a**,**b**) FE-SEM images of the ZnO/ZnS core–shell/CNT nanostructures synthesized with sulfurization periods of (**a**) 15 and (**b**) 30 min; (**c**,**d**) SEM cross-section images with a sulfurization time of (**c**) 15 and (**d**) 30 min.

**Figure 3 nanomaterials-10-01521-f003:**
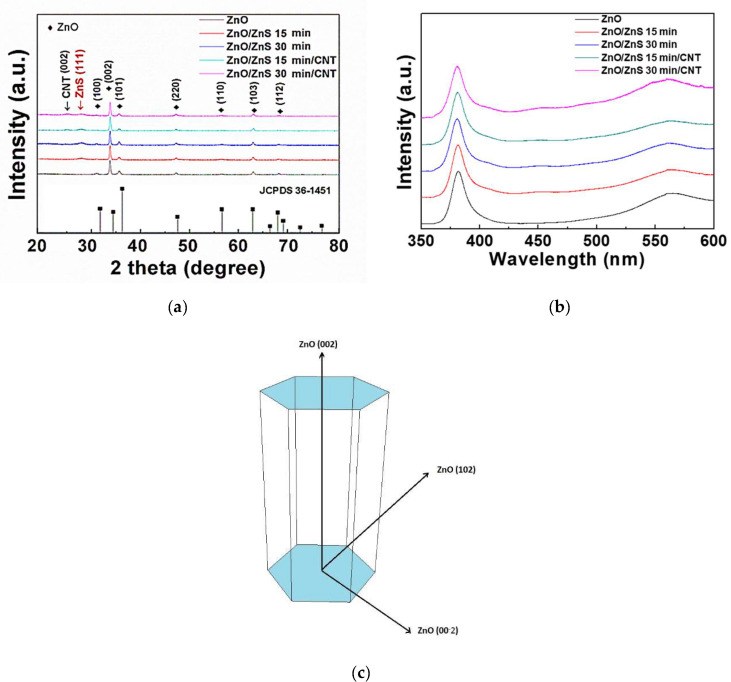
(**a**) XRD patterns of ZnO/ZnS core–shell structures and ZnO/ZnS/CNT nanocomposites; (**b**) photoluminescence (PL) measurements of ZnO/ZnS core–shell structures and ZnO/ZnS/CNT nanocomposites with various sulfurization periods; (**c**) Several orientations of hexagonal zinc oxide nanorods.

**Figure 4 nanomaterials-10-01521-f004:**
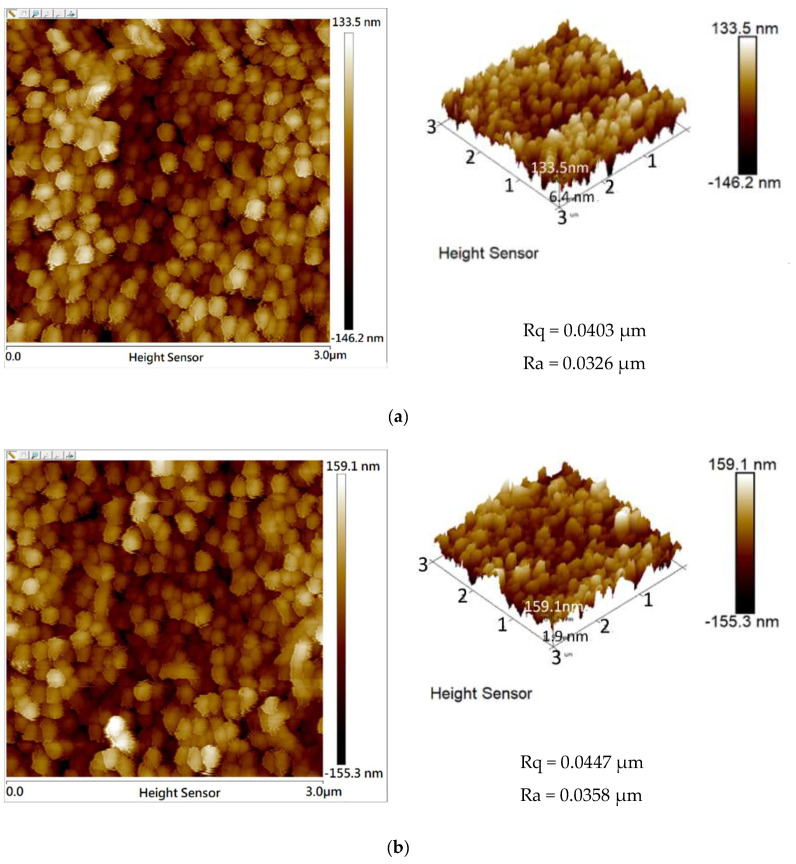
Atomic force microscopy (AFM) images of ZnO/ZnS core–shell/CNT structures with sulfurization times of (**a**) 15 min, the horizontal scale bar is presented from 0.0 μm to 3.0 μm and the perpendicular scale bar is presented from 133.5 nm to −146.2 nm and (**b**) 30 min, the horizontal scale bar is presented from 0.0 μm to 3.0 μm and the perpendicular scale bar is presented from 159.1 nm to −155.3 nm.

**Figure 5 nanomaterials-10-01521-f005:**
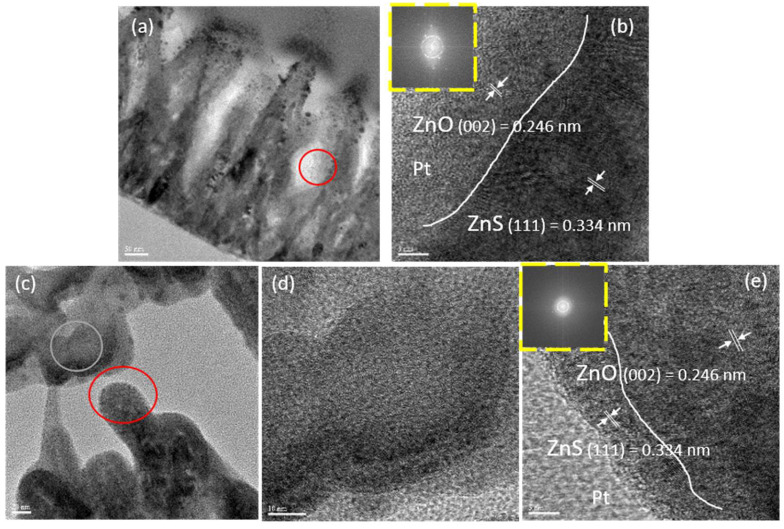
(**a**) A ZnO/ZnS core–shell image, scale bar = 50 nm; (**b**) a HRTEM image of ZnO/ZnS border, scale bar = 5 nm, and a digital microscope-processed image of the ZnO/ZnS interface; (**c**) a TEM image with CNTs (gray circle) and without CNTs (red circle), scale bar = 20 nm; (**d**) a HRTEM image on the grey circle, scale bar = 10 nm; (**e**) a HRTEM image on the red circle, scale bar = 5 nm, and a digital microscope-processed image of the ZnO/ZnS interface.

**Figure 6 nanomaterials-10-01521-f006:**
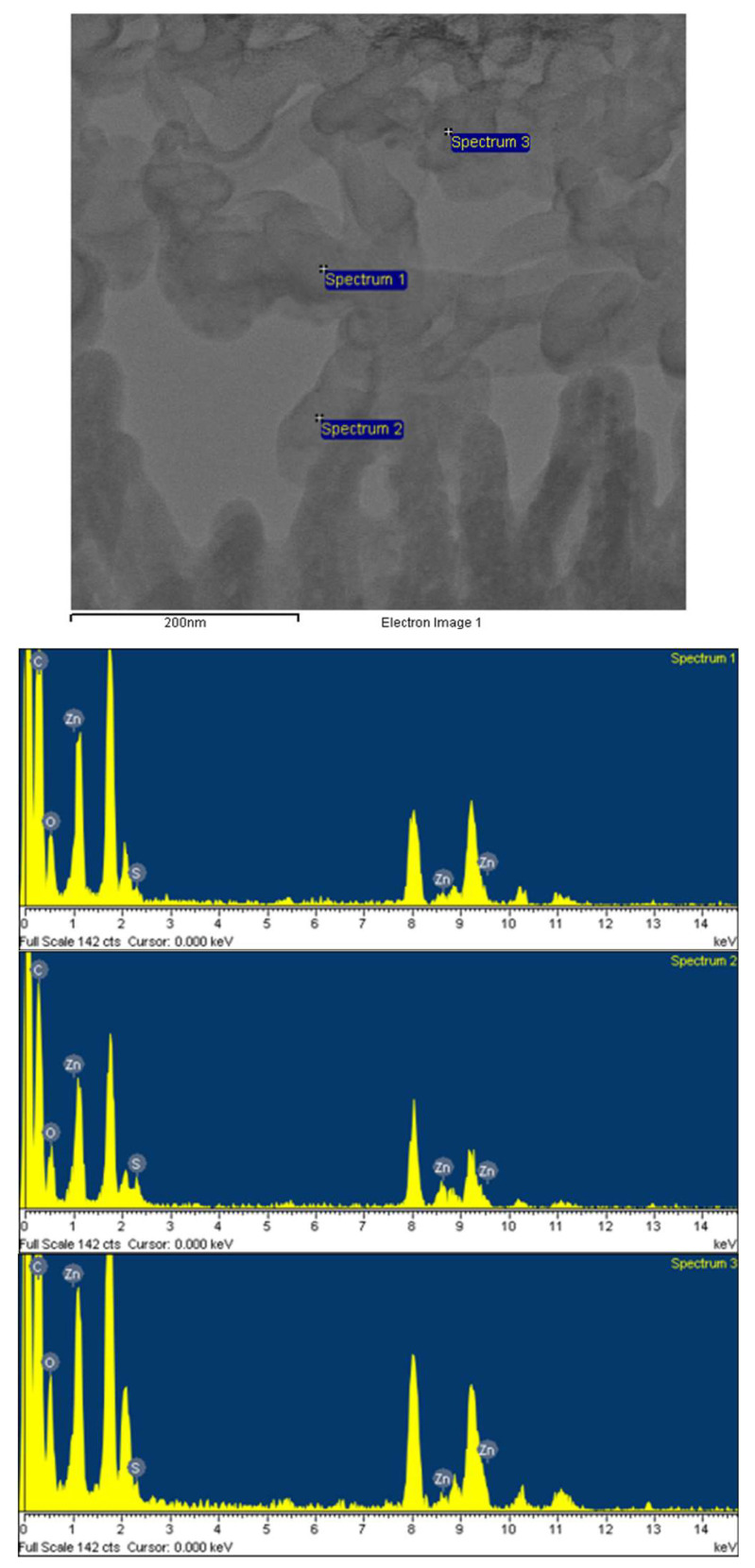
EDS analysis of the ZnO/ZnS core–shell/CNT nanostructures.

**Figure 7 nanomaterials-10-01521-f007:**
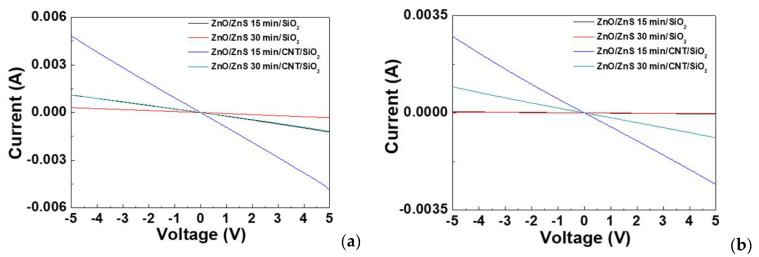
Current–voltage curve (I–V curve) for the ZnO/ZnS 15 min/SiO_2_-based light sensor, ZnO/ZnS 30 min/SiO_2_-based light sensor, ZnO/ZnS 15 min/CNT/SiO_2_-based light sensor, and ZnO/ZnS 30 min/CNT/SiO_2_-based light sensor (**a**) in the dark and (**b**) under the light.

**Table 1 nanomaterials-10-01521-t001:** EDS analysis of the ZnO/ZnS core–shell/CNT nanostructures.

Locations	Weight %				Atomic %			
	Zn	O	S	C	Zn	O	S	C
Spectrum 1	0.16	15.34	0.84	83.65	0.03	12.06	0.33	87.58
Spectrum 2	6.24	15.59	2.89	75.28	1.29	13.12	1.21	84.38
Spectrum 3	0.01	16.08	0	84.05	0.00	12.56	0	87.49

**Table 2 nanomaterials-10-01521-t002:** Comparison of dark current and light current at a fixed voltage of −2 V toward the samples of ZnO/ZnS 15 min, ZnO/ZnS 30 min, ZnO/ZnS 15 min/CNTs, and ZnO/ZnS 30 min/CNTs.

	Voltage	Dark Current	Light Current
ZnO/ZnS 15 min	−2 V	1.69914 × 10^−2^ mA	4.51376 × 10^−1^ mA
ZnO/ZnS 30 min	−2 V	9.72357 × 10^−3^ mA	1.25697 × 10^−1^ mA
ZnO/ZnS 15 min/CNTs	−2 V	1.03 mA	1.87 mA
ZnO/ZnS 30 min/CNTs	−2 V	3.5489 × 10^−1^ mA	4.34431 × 10^−1^ mA

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
