# Peer review of "Morphological, Material, and Optical Properties of ZnO/ZnS/CNTs Nanocomposites on SiO_2_ Substrate"

_nanomaterials, 2020, doi:10.3390/nano10081521_

Round 1
Reviewer 1 Report
The modified manuscript is OK.
Reviewer 2 Report
The authors have successfully answered most of the questions and I think that the overall quality of the manuscript has improved and it can be published in the present form.
This manuscript is a resubmission of an earlier submission. The following is a list of the peer review reports and author responses from that submission.
Round 1
Reviewer 1 Report
The manuscript "Morphological, material and optical properties of ZnO/ZnS/CNTs nanocomposites on SiO2 substrate" can be of potential interest to the readership of Nanomaterials even though this subject has already benn extensiely studied, but major revisions are needed for the possiblity to be published.
Firstly, I will concentrate on the structural part (especially XRD study) of the manuscript:
- Line 106, XRD stands for X-ray diffraction, not X-ray diffractometer, I believe it is known even to first year students of any materials course.
- Line 107, the authors state that XRD patterns are shown in Fig. 3 (a), and in the manuscript the Figure 3 (a) shows PL measurements.
- Line 108, JCPDS stands for Joint Commitee on Powder Diffraction Standards not Powder Diffusing Standards Joint Commitee. This mistake is truly not understandable, anybody with any experience with XRD would not write something like that.
- Line 111, what is the meaning of crystalline phase (002)? 002 describes lattice planes of the phase, not the phase itself.
- In the Fig. 3 (b) why the XRD patterns of pure ZnO are not shown?
- All the patterns in the Fig. 3 (b) look the same, what is the difference between them? Mainly, concerning the composition regarding ZnS phase and CNT phase?
- identification using the JCPDS cards is really not acceptable in the manuscript where the structure of investigated materials is a major part. Rietveld refinement of the XRD patterns, including full qualitative (phase identification) and quantitative (amounts of each phase), as well as microstrucural (size-strain analysis) is needed.
- lines 129-133, what is the meaning of surface roughness? What is the impact on the properties of the material?
- Why there is no electron difraction images? It could be helpfull in the structure analysis of the studied materials.
- Lines 168-171, what is the meaning of top (lower Zn concentration) and bottom (higher Zn concentration)?
- Why the EDS analysis wasn't performed on all the samples?
- I still don't understand what is the real-life advantage of these materials,
Reviewer 2 Report
The author should prepare this manuscript more carefully. For example, it was first said silicon substrates were used in this study (page 2, line 47), then said SiO2 substrates were used (page 2, line 57); in Fig. 3 (page 5), Figs (a) and (b) are misplaced; and which one is (a), which is (b) in Fig 7 (page 8)?
Also, it was said the AFM measured sample surface roughness (Ra) results were 0.0326 nm and 0.0358 nm, respectively (page 5, line 131), however, the unit here should be micrometer (um), not nanometer (nm)!
The authors claimed that the samples with CNTs had strong defect luminescence (page 4, line 123), however, the PL results (Fig. 3) didn’t show obvious differences among the different samples to support this claim.